# Satellite-Aided Consensus Protocol for Scalable Blockchains

**DOI:** 10.3390/s20195616

**Published:** 2020-10-01

**Authors:** Xintong Ling, Zheng Gao, Yuwei Le, Li You, Jiaheng Wang, Zhi Ding, Xiqi Gao

**Affiliations:** 1National Mobile Communications Research Laboratory, Southeast University, Nanjing 210096, China; xtling@seu.edu.cn (X.L.); zgao@seu.edu.cn (Z.G.); ywle@seu.edu.cn (Y.L.); liyou@seu.edu.cn (L.Y.); xqgao@seu.edu.cn (X.G.); 2The Purple Mountain Laboratories, Nanjing 210023, China; 3Department of Electrical and Computer Engineering, University of California, Davis, CA 95616 USA; zding@ucdavis.edu

**Keywords:** blockchain, consensus protocol, performance analysis, satellite, space–terrestrial network

## Abstract

In this work, we propose a satellite-aided permissionless consensus protocol for scalable space–terrestrial blockchains. We design its working principle and workflow by taking full advantage of satellites for extensive coverage and ubiquitous connectivity. Based on the proposed protocol, we demonstrate how such a space–terrestrial blockchain grows and evolves through several typical cases in the presence of adversarial nodes, user misbehavior, and transmission outage. Taking proof of work (PoW) as a benchmark, we assess the system security by considering both adversarial miners and possible colluding satellites. Then, we analyze the maximum blockchain throughput under network capacity limits and evaluate the impact of information propagation delay via a Markov model. Simulation results support that the proposed satellite-aided consensus protocol achieves higher throughput and exhibits greater scalability than PoW.

## 1. Introduction

Blockchain is currently blossoming in sectors such as internet of things (IoT) [1,2], 6G wireless networking [3,4], edge computing [5,6], sensor networks [7], and smart city [8]. For such a secured distributed ledger technology, consensus protocols serve as the very foundation. Based on specific consensus mechanisms, blockchain spurs miners to participate in maintaining an ordered public ledger via proper financial incentives. PoW, the most popular and recognized consensus protocol, was first introduced in 1992 [9] and developing into a vital part of Bitcoin since its launch in 2008 [10]. The success of PoW-based cryptocurrencies in the last decade indicates its robustness and resilience against misbehaviors and adversarial attacks [10,11].

However, PoW has an obvious weakness, as it consumes immense amounts of resources and energy. The estimated energy consumed by Bitcoin was around 70.05 TWh in 2020 [12], equivalent to the annual electricity consumption of the entire country of Czech Republic. Alternatives to PoW leverage other capabilities instead of intensive computations but often cause new security concerns [13]. Constrained by the consensus protocol (as well as the chain structure), the existing public blockchains exhibit scalability problems. Transactions per second (TPS), as the measure of information throughput in blockchain, is rather low in traditional blockchains, e.g., at most 7 TPS in Bitcoin and 15 TPS in Ethereum [14]. As a benchmark, centralized entities such as PayPal and VISA can achieve hundreds or thousands of TPS [15]. In short, high consumption and low scalability present two main barriers that severely constrain the advances of blockchain technologies.

Currently, there are several works concerned with how to utilize blockchains for the space sector [16,17,18], such as mobile satellite communication networks (MSNET) [16] and satellite monitoring [17]. However, only a few works and projects were investigating the potential applicability of satellites on blockchain. In both of [19,20], the authors proposed to use satellites as relays to help forward and propagate blocks in a space–terrestrial network. The project of BitSats [21] planned to turn satellites into crucial full Bitcoin nodes in space as early as 2014. Another project named SpaceChain [22] aims to bring an open-source decentralized software model to space and has leased two satellites in February and October of 2018, respectively. More recently, BlockStream [23] a startup in Canada, has launched six satellites to broadcast full Bitcoin blockchain for air-gapped areas across the globe. Another project ChainSat [24] has a similar plan to facilitate an off-grid network for enterprise blockchain ledgers via direct satellite communications. Space Impulse [25] and SupremeSat [26] also are reported to introduce blockchains into the space industry.

Notably, satellites can bring substantial benefits to next-generation blockchain protocols because of the extensive coverage and ubiquitous connectivity. As standardized by, e.g., Digital Video Broadcasting Second Generation Satellite (DVB-S2) and Digital Video Broadcasting Second Generation Satellite Extensions (DVB-S2X) [27,28], the satellites multicast transmission is now mature enough to deliver information to massive terrestrial receivers simultaneously, which would be helpful to solve the scalability issue of blockchain. (We will show how multicast transmissions affect the blockchain scalability in Section 5). Moreover, the latency of the terrestrial peer to peer (P2P) networks (e.g., the Internet) is often long-tailed due to the number of hops [29], whereas the propagation delay of satellite communications is nearly fixed and more controllable [19,20]. Furthermore, satellite systems can significantly reduce the vulnerability against Internet attacks and network interrupts [30,31]. Satellites are becoming the next game changer for blockchains.

However, the existing space blockchain solutions have not fully considered the characteristics of satellite communications, such as delay, link budget, bandwidth, on-board resources [16,32,33]. These solutions rely on the conventional, costly mining such as PoW, and use satellites merely as relays to forward blocks for unserved and under-served areas (e.g., BlockStream and ChainSat), or accelerate information diffusion in the mining network (e.g., [19,20]). Especially, if satellites are deployed as full blockchain nodes (e.g., BitSat and SpaceChain), block verification and chain backup put extra stress on the limited on-board device process and storage capability, which may explain why the project BitSat stalled eventually. Moreover, the uplink is dilemmatic. The uplink has to go through specific authorized ground stations [20], e.g., Blockstream only announced the availability of one-way satellite broadcasting. Ordinary users cannot upload newly mined blocks directly, which actually hurts the decentralization of the blockchain system. Nevertheless, even if ordinary users can upload blocks, high-speed uplinks are required to reduce the transmission delay, which, however, would impose extra cost on ordinary miners and raises the barriers to entry. Even though all these above issues are tolerated within the Bitcoin blockchain, they would become bottlenecks for next-generation high-TPS blockchains in the future.

Therefore, by comprehensively taking the characteristics of satellites into account, we propose a satellite-aided consensus protocol for efficient space–terrestrial blockchains. Our proposed protocol only needs low-speed, one-way multicast transmissions from space to terrestrial, which not only addresses the uplink dilemma but also vastly reduces the transmission delay. By leveraging the features of satellite multicast transmissions, the proposed protocol does not rely on the costly PoW protocol, and is thus eco-friendly and sustainable without consuming tremendous energy on meaningless hash queries. The main contributions of our work include:We develop a satellite-aided permissionless consensus protocol for sustainable, scalable blockchains, which, to the best of our knowledge, has never been investigated before.Technically, we demonstrate how the satellite-aided blockchain evolves and eventually converges despite various misbehaviors on a case-by-case basis, and show how the transmission outage and possible adversarial participants affect the chain evolution.Using the PoW-based blockchain as a benchmark, we theoretically analyze and assess the system performance regarding security and network throughput.Simulation results demonstrate that the satellite-aided blockchain exhibits greater scalability and can achieve higher TPS than the traditional PoW-based blockchains.

This paper is organized as follows. Section 2 presents the principle and workflow of the satellite-aided consensus protocol. Section 3 illustrates the possible patterns of blockchain evolution. Section 4 and Section 5 calculate the blockchain performance in terms of security and scalability, respectively. At last, Section 6 provides the simulation results, and the paper concludes with Section 7.

## 2. Satellite-Aided Consensus Protocol

There are essentially three types of orbits classified by the satellite altitude: geostationary earth orbit (GEO), medium earth orbit (MEO), and low earth orbit (LEO) [32,33,34]. Among them, GEO satellites are stationary relative to the earth’s surface so that the doppler shift is negligible and has a lower transmission outage probability than non-GEO satellites [32,35,36]. The GEO satellites work at very high altitudes (≈35,786 km) and can offer the most extensive coverage [37]. Thanks to the low outage probability and wide coverage, GEO satellites are preferred in our proposed protocol. However, the high altitudes of GEO satellites also result in long delays and high path losses [32,35]. Thus, we should design the consensus protocol by taking the drawbacks of GEO into account.

The brief workflow of the satellite-aided consensus protocol is illustrated in Figure 1. The proposed protocol requires low-speed, one-way transmissions only, which effectively reduces the negative impact of round-trip delays. In each round, a GEO satellite periodically multicasts random oracles (typically of hundreds of bits) to the on-ground terminals based on the standards such as DVB-S2 and DVB-S2X [27,28].

These random oracles are digitally signed by the satellite to avoid forgery and required to be included in the generated block for validation. The on-ground miners only need to capture random oracles from space through terrestrial user terminals, such as portable mobile receivers, very small aperture terminals (VSATs), among others [32]. A random oracle will point to a specific terrestrial miner and elect it as a winner of the current round. The winner is granted an opportunity for proposing a new block on the top of the existing chain and spreading it among terrestrial miners. Via satellite multicasting (and also possible terrestrial forwarding), the majority of terrestrial miners receive the random oracle and thus can verify the identity of the winner and validate the corresponding block. Therefore, the satellite-aided blockchain grows as one block is built on the top of others.

We design the above workflow by comprehensively considering the characteristics of GEO satellites. The benefits of our proposed satellite-aided protocol are hence multifold. First, the protocol imposes a minimal burden on satellites with no need to process, store, verify, or even forward blocks in space. Second, since the random oracles are generated by satellites, the uplinks are unnecessary in the proposed protocol, which reduces the delay caused by GEO, solves the uplink dilemma, and further simplifies the space–terrestrial network structure. Third, we only multicast random oracles (hundreds of bits) instead of the whole blocks (typically of megabytes). On the one hand, it can reduce the influence of high path loss of GEO satellites, letting more miners at a low signal-to-noise ratio (SNR) capture the oracles. On the other hand, it lowers the link budget and is scalable for high TPS blockchains. Fourth, our solution does not require miners to perform meaningless hash queries but takes advantage of multicast transmissions to maintain the system consensus in a novel approach, which largely reduces the mining cost and is thus sustainable and green. By leveraging the characteristics of GEO satellites including its extensive coverage and low outage probability, the satellite-aided blockchain is more scalable and can achieve a higher TPS than traditional blockchains. (Please see the analysis in Section 5).

In the satellite-aided consensus protocol, random oracles can be generated based on real-time physical quantities measured by the satellite, such as cosmic rays, hydromagnetic waves, transient radiations, and so forth [38,39,40]. Hence, the random oracle can be truly random rather than pseudo-random to guarantee mining fairness. However, the corresponding services have to be ordered and purchased from satellite operators to generate and multicast random oracles from the space, which increases the maintenance cost. In fact, there is another tricky approach to obtain oracles. The random oracle can be generated by data packets multicast to the ground for other purposes such as satellite television (TV) and global positioning system (GPS). The consensus protocol gives a pre-defined rule to determine the specific packets (e.g., satellite sources and frequency bands) for oracle generation in each round. In this case, the satellite even do not realize that they participate in generating oracles and maintaining a blockchain, and it is thus unnecessary to purchase any satellite services. However, in this manner, the oracle randomness may be affected and the oracles are more likely to be manipulated by adversaries. Therefore, there exists a trade-off between security and cost in the oracle generation, which should be selected according to the purposes and requirements of blockchain.

Note that it might be risky to directly select winners by random oracles in fear of Sybil attack–malicious users can create many fake identities to increase the winning probability. Hence, we can draw the principle from proof of stake (PoS). The oracle can be interpreted as an index in the list of all the already minted coins so that its owner is the winner of this round [13]. The winning probability is only related to the stake that miners hold and does not increase by creating multiple identities. Therefore, Sybil attacks can be prevented effectively, and satellites do not have to know or maintain a list of legitimate miners.

## 3. Patterns of Chain Evolution

In this section, we show how such a satellite-aided blockchain extends and evolves. As shown in Figure 2a, an honest winner $w_{2}$, after being chosen by the oracle in round $r_{2}$, would generate a valid block $b_{2}$ on the top of the last block $b_{1}$ and multicast it to the rest of the mining network. According to the oracle of $r_{2}$, other miners can verify the identity of $w_{2}$ and the validation of $b_{2}$. Usually, the next winner $w_{3}$ would propose a new block $b_{3}$ on the top of $b_{2}$. As such, the chain will grow and evolve.

However, due to network delays or misbehaviors, exceptions may occur. As shown in Figure 2b, it is possible that the chosen winner $w_{2}$ misses the oracle due to the outage of satellite links, and thus does not generate any new block in round $r_{2}$. As a result, the winner $w_{3}$ of the next round will generate a block $b_{3}$ on the top of $b_{1}$ directly. While in Figure 2c, $w_{3}$ may not receive $b_{2}$ in round $r_{3}$ due to network delay. Since $w_{3}$ has not been informed, it generates a new block $b_{3}$ on the top of $b_{1}$ directly. However, in this case, $b_{2}$ exists at the same height, resulting in orphaned blocks. The future block extension will determine which block will be finally included in the main chain.

In terms of misbehaviors, a winner $w_{2}$ in Figure 2d may generate an invalid block, whereas, in Figure 2e, a miner $w_{2}^{\prime }$ who has not been selected, may forge its identity and generate a block $b_{2}^{\prime }$. In the above two cases, other miners can easily find that the block in Figure 2d is invalid and the block in Figure 2e is generated by an illegitimate winner by checking the block and oracle. Hence, these two $b_{2}^{\prime }$ in Figure 2d,e will be ignored and excluded from the main chain. Furthermore, in Figure 2f, a malicious winner $w_{2}$ may diverge the chain by publishing two or more valid blocks in one single round. As a rule of thumb, benign miners should discard all of them.

Last but not least, the alternative history attack is likely to happen in a satellite-aided blockchain, and could bring disastrous effects to the blockchain systems. When an alternative history attack takes place, an adversary would create a fraudulent branch and secretly mine on it, as shown in Figure 2g. Until the adversarial branch is longer (or heavier) than the original benign one, the adversary will release it to alter a confirmed history. The probability of a successful alternative history attack, also known as the confirmation error probability [11], is thus used as a measure of chain security in this paper.

In a nutshell, Figure 2a is the ideal pattern for blockchain extension, as one block is generated on the top of the chain in each round. Figure 2b,c are caused by the oracle multicast outage and block propagation delay and are thus inevitable in practice. Orphaned blocks may occur in these two cases and their combinations. The possible misbehaviors in Figure 2e,f can be effectively prevented by the proposed protocol. Essentially, the attack in Figure 2g poses an inherent risk of a distributed system, and hence the successful attack probability will be discussed in the following section to assess the resilience of the proposed protocol.

## 4. Security Implications

In this section, we assess the risk to the satellite-aided consensus protocol in the presence of an adversary. Denote $\beta_{t}$ as the fraction of the total terrestrial mining power controlled by an adversary. Hence, in a round, the probability that a benign miner is chosen as the winner is $1-\beta_{t}$, and the winning probability of an adversary is $\beta_{t}$. Moreover, we have to consider that some satellites may be controlled or hacked by an adversary. Let $\beta_{s}$ be the fraction of adversarial satellites. Undoubtedly, when an adversarial satellite generates an oracle, an adversarial miner will be selected as the winner. In short, the probability that a malicious miner is selected as the winner is $\beta_{s}+\beta_{t}\left(1-\beta_{s}\right)$, while the probability that a benign miner is selected as the winner is $\left(1-\beta_{s}\right)\left(1-\beta_{t}\right)$.

More practically, we should take the possible outage or interrupt of satellite links into considerations. Denote *p* as the successful transmission probability of satellite links such that the outage probability is 1−p. Figure 2b shows that a winner will not generate a block if it does not receive the oracle (with probability 1−p). Hence, the growth rate of the benign and adversarial chains can be, respectively, formulated as
(1)$$\begin{array}{cc} h & =p\left(1-\beta_{s}\right)\left(1-\beta_{t}\right), \end{array}$$
(2)$$\begin{array}{cc} f & =p\left(\beta_{s}+\beta_{t}-\beta_{t}\beta_{s}\right). \end{array}$$

Therefore, the overall relative power controlled by an adversary, denoted by β, can be expressed by
(3)$$\beta ≜\frac{f}{f+h}=\beta_{s}+\beta_{t}-\beta_{t}\beta_{s}.$$

From Equation (Equation 3), the successful transmission probability *p* will not directly influence the system security. However, if the adversarial miners have back channels to the colluding satellites, the blockchain faces heightened risk. In the absence of adversarial satellites, the overall adversarial power only depends on the terrestrial miners $\beta =\beta_{t}$ as $\beta_{s}=0$. Obviously, the existence of adversarial satellites makes the chain more vulnerable.

We use the probability of a successful double spending attack, i.e., confirmation error probability, to evaluate the resilience of the proposed protocol. According to the framework on alternative history attacks [41], given the confirmation number *k*, the confirmation error probability is
(4)Skβ=1−∑n=0kn+k−1nβn1−βk−βk1−βnif β<1/21if β≥1/2.

**Remark** **1.**
*For β≥1/2, the adversary dominates the chain and always succeeds. For β<1/2, the confirmation error probability is exponentially decreasing with the confirmation number k, where the decay rate is the attacker’s relative power β. Note that a PoW-based blockchain with adversary mining power $\beta_{PoW}=\beta$ has the exact same confirmation error probability as a satellite-aided blockchain.*


## 5. Throughput Analysis

### 5.1. Network Capacity Limits

In the context of blockchain, throughput means the amount of information, e.g., transactions, gets confirmed per second. We would like to characterize the maximum throughput of the satellite-aided consensus protocol. Let us say the maximum block size limit is *B* bits, and each round corresponds to Δ seconds. Hence, since at most one valid block can be generated in each round, the maximum throughput is B/Δ bps. However, the presence of the adversary and the outage of satellite links degrade the blockchain throughput. Based on the growth rate of the benign chain in Equation (Equation 1), the throughput of the satellite-aided consensus is
(5)$$T_{\mathrm{CPTY}}=\frac{hB}{\Delta }=p\left(1-\beta \right)\frac{B}{\Delta }.$$

Let *C*(bps) be the capacity of the terrestrial mining network. Obviously, it is impossible to deliver a block with size *B* within the interval Δ, if B/Δ>C. Hence, the values of *B* and Δ are constrained by B/Δ≤C. The maximum throughput of a satellite-aided consensus under capacity limits is given by
(6)$$T_{\mathrm{CPTY}}\leq p\left(1-\beta \right)C≜T_{\mathrm{CPTY}}^{m}.$$

As a benchmark, we present the throughput of PoW-based consensus borrowed from a recent work [11]. In PoW, the block generation obeys a Poisson process with mean λ, wherein β fraction of miners are adversarial and the rest are benign. During the period τ, the probability of no benign block is $\exp\left(-\lambda (1-\beta )\tau \right)$. Hence, the growth rate of the benign chain (at least one benign block during τ) is $1-\exp\left(-\lambda (1-\beta )\tau \right)$, and the throughput is
(7)$$T_{\mathrm{PoW}}=\left(1-\exp\left(-\lambda \left(1-\beta \right)\tau \right)\right)B/\tau .$$

As the block generation rate λ cannot exceed the limit C/B, the limiting throughput $T_{\mathrm{PoW}}$ is thus bounded by
$$\begin{array}{cc} T_{\mathrm{PoW}} & \leq \left(1-\exp\left(-\frac{\tau C}{B}\left(1-\beta \right)\right)\right)\frac{B}{\tau } \end{array}$$
(8)$$\begin{array}{c} \leq \left(1-\exp\left(\beta -1\right)\right)C≜T_{\mathrm{PoW}}^{m}. \end{array}$$

The equality is achieved if and only if λ=C/B=1/τ.

By comparing $T_{\mathrm{CPTY}}^{m}$ in Equation (Equation 6) with $T_{\mathrm{PoW}}^{m}$ in Equation (Equation 8), we have $T_{\mathrm{CPTY}}^{m}>T_{\mathrm{PoW}}^{m}$ if $p\left(1-\beta \right)\geq 1-\exp\left(\beta -1\right)$. We thus define $y\left(\beta \right)≜\frac{1-\exp\left(\beta -1\right)}{1-\beta }$ as the threshold of *p*. If the quality of multicast channel is better than a given threshold related to β, i.e., $p>y\left(\beta \right)$, then the satellite-aided consensus protocol has a larger maximum throughput than PoW.

Note that $y\left(\beta \right)$ is monotonically increasing in β within 0≤β<1. To show this, for any β<1, we have
$$y^{\prime }\left(\beta \right)=\frac{\left(\beta -2\right)\exp\left(\beta -1\right)+1}{{\left(\beta -1\right)}^{2}}>\frac{-1+1}{{\left(\beta -1\right)}^{2}}=0,$$
where the inequality is because $\left(\beta -2\right)\exp\left(\beta -1\right)$ is lower bounded by $\left(\beta -2\right)\exp\left(\beta -1\right){|}_{\beta =1}=-1$. As a result, for any β≤0.5, if $p>y\left(\beta =0.5\right)\approx 0.787$, then the proposed consensus protocol has a higher throughput than PoW as $T_{\mathrm{CPTY}}^{m}>T_{\mathrm{PoW}}^{m}$ always holds. As a benchmark, the outage probability of China’s Beidou navigation satellite system is about 2% [42], i.e., p≈98%, which is much higher than the threshold $y\left(\beta =0.5\right)$. If $p<y\left(\beta =0\right)=1-\exp\left(-1\right)$, then $T_{\mathrm{CPTY}}^{m}<T_{\mathrm{PoW}}^{m}$.

**Remark** **2.**
*The above conclusion implies that the scalability of satellite-aided blockchains is highly related to the quality of the satellite multicast channels. Essentially, high-quality multicast transmissions change the information flow of the satellite-aided blockchain, thereby significantly improving the throughput. As indicated by the above discussions, if the multicast channel is in poor quality, the satellite-aided consensus protocol cannot benefit from the nature of multicast transmissions. Furthermore, since the threshold $y\left(\beta \right)$ grows with β, better multicast channels with a larger p are thus required if the adversary is more powerful.*


### 5.2. Propagation Delay

In the above analysis, we considered the capacity limits of B/Δ≤C. As shown in [29], the spreading delay across, e.g., 90% nodes, via Internet could be quite long. Hence, we define i(t) as the ratio of informed nodes at time *t* for a specific network. Take t=Δ as an example, where the round slot is Δ seconds. Hence, $i_{\Delta }≜i\left(t=\Delta \right)$ is the ratio of nodes receiving the block in the last round. After the block of last round is published, $1-i_{\Delta }$ fraction of nodes will not receive that block in the next round, potentially leading to orphaned blocks as the case in Figure 2c. However, such orphaned blocks lower the chain growth rate and also affect the blockchain throughput, even without an adversary. (As a rule of thumb, if two or more forks occur, the next winner will always follow the longest known chain. If these forks are at the same height, the next winner will follow the one which achieves this height earlier).

Now things get complicated. In Figure 3a, starting from a consistent state, the number of possible blockchain states grows exponentially after just three rounds. Note that the ratio of informed nodes after one round, i.e., $i_{\Delta }$, is the most critical term affecting orphaning. To characterize this complicated problem, we focus on the factor $i_{\Delta }$ and assume $i\left(k\Delta \right)=1$ for k=2,3,... without affecting the spirit of our results. These countless states can be classified into four categories. We can abstract their states via the minimum height gap between the longest chain and other active forks. In Figure 3a, we start from a consistent state, denoted as $S_{\infty }$, since there exists a unique active chain. If the next winner receives the last block, it is still in state $S_{\infty }$. Otherwise, it jumps to the state $S_{0}$ with a newly-generated fork. The next block generated on any fork will lead to a longer main chain by one block, i.e., state $S_{1}$. Then, in the next round, if the winner follows the longest branch, the blockchain returns to a consistent state $S_{\infty }$, otherwise it jumps back to the tied state $S_{0}$. For any state, if the next winner misses the oracle or is an adversary, no block is generated or released in this round, and the chain goes to the state $S_{\times }$.

One can see that these four states form a Markov chain, as shown in Figure 3b. We can mathematically write the probability transition matrix as follows:$$Q=\left(\begin{array}{ccccc} S_{0}: & 0 & h & 0 & 1-h\\ S_{1}: & h\left(1-i_{\Delta }\right) & 0 & hi_{\Delta } & 1-h\\ S_{\infty }: & h\left(1-i_{\Delta }\right) & 0 & hi_{\Delta } & 1-h\\ S_{\times }: & 0 & 0 & h & 1-h \end{array}\right),$$
where *h* is the growth rate of the benign chain defined in Equation (Equation 1). Let w be the steady-state distribution of the considered Markov chain. The expression of w can be obtained from wQ=w:$$w=\left(\begin{array}{cccc} \frac{h^{2}\left(1-i_{\Delta }\right)}{\left(1-i_{\Delta }\right)h+1}| & \frac{h^{3}\left(1-i_{\Delta }\right)}{\left(1-i_{\Delta }\right)h+1} & |\frac{h-h^{3}\left(1-i_{\Delta }\right)}{\left(1-i_{\Delta }\right)h+1} & |1-h \end{array}\right).$$

Note that, these four states have different probabilities to extend a new block in the longest chain. By comprehensively considering these states, we can obtain the growth rate:$$h_{\mathrm{PROP}}=w_{0}h+w_{1}hi_{\Delta }+w_{\infty }hi_{\Delta }+w_{\times }h=\frac{h}{1+\left(1-i_{\Delta }\right)h}.$$

As a result, the blockchain throughput with information propagation delay can be expressed as
(9)$$T_{\mathrm{PROP}}=\frac{h_{\mathrm{PROP}}B}{\Delta }\leq \frac{h}{1+\left(1-i_{\Delta }\right)h}C≜T_{\mathrm{PROP}}^{m}.$$

**Remark** **3.**
*Comparing Equations *(Equation 6)* and *(Equation 9)*, we can find the impact of information propagation on blockchain throughput. Only considering at most two-round propagation delay, we can see the maximum throughput degradation by a factor of $1/\left(1+\left(1-i_{\Delta }\right)h\right)$. In principle, we cannot control the value of $i_{\Delta }$ for a given network but can select the proper block size B and the round slot *Δ* based on *(Equation 9)* to maximize the blockchain throughput.*


Usually, propagation delays do not affect the behaviors of colluding adversarial miners so that the growth rate of the adversarial chain will be the same as Equation (2). Therefore, propagation delay not only affects the throughput but also may weaken the double-spending resilience.

## 6. Evaluation

In this section, we evaluate the proposed satellite-aided consensus protocol from throughput and security aspects. We establish a satellite-aided blockchain simulation system according to the proposed protocol. (Please see github.com/xtling/SatelliteChain for the source code.) In our setting, terrestrial miners form a peer-to-peer network with the same winning probability and the same transmission outage from space. In Figure 4, Figure 5 and Figure 6, simulation results are shown by markers, whereas analytical curves are illustrated by dashed lines. We require k=6 blocks to confirm a block, and normalize the throughput by the mining network capacity *C*.

In Figure 4, we illustrate the throughput and security of the proposed protocol in the presence of one adversary. One can see that the simulation results (markers) match our analytical throughput and security results (dashed lines). As shown in Figure 4, both space and terrestrial safety levels ($\beta_{s}$ and $\beta_{t}$) compromise the security of the whole blockchain. However, to manipulate the blockchain, an adversary must control a vast number of resources $\beta =\beta_{s}+\beta_{t}-\beta_{t}\beta_{s}>\frac{1}{2}$ in the mining network.

In Figure 5, we compare the proposed satellite-aided consensus protocol with PoW as a benchmark. The throughput and security properties of PoW have been presented in [11,41], respectively. The proposed consensus protocol can achieve much higher throughput than PoW with comparable safety property as PoW. Moreover, one can see that the throughput of the satellite-aided blockchain depends heavily on the quality of satellite multicast channels, which is consistent with the remarks in Section 5.1.

In Figure 6, we demonstrate the impact of network propagation factor $i_{\Delta }$ on blockchain throughput. The analytical results (lines) based on the Markov model is closely matched by simulations (markers). As shown in Figure 6, the network propagation delay leads to more orphaned blocks and degrades the network throughput by a factor $1/\left(1+\left(1-i_{\Delta }\right)h\right)$ shown in Equation (Equation 9). As a guideline for practical design, for a given network with i(t), we must systematically optimize the block size *B* and round slot Δ to obtain a proper i(Δ) for higher throughput.

## 7. Conclusions and Future Research

Utilizing satellite systems for advanced public blockchains is a promising approach, due to ubiquitous connectivity and wide coverage. Hence, this study proposed a novel satellite-aided consensus protocol for efficient space–terrestrial blockchains. Unlike existing works [19,20], we used satellites as neither full blockchain nodes nor relays. In our study, we do not rely on costly consensus protocols such as PoW but designed a proper working principle according to the space–terrestrial networks’ characteristics. We analyzed the system performance regarding throughput and security. In terms of security, we took both adversarial miners and satellites into consideration. In the throughput analysis, we investigated the maximum throughput under the capacity limits and further modeled the case with propagation delays by a Markov chain. Compared with the traditional PoW, the proposed consensus protocol exhibits higher scalability.

Remark that satellite-aided blockchains also open several interesting future directions.

Access control. As pointed out in Section 4, a larger fraction of adversarial satellites $\beta_{s}$ will increase the confirmation error probability, which means an adversary can make the blockchain more vulnerable by hacking satellites. Therefore, access control is vital to the space–terrestrial blockchains [43]. On the one hand, our protocol does not require (or allow) terrestrial networks to upload data to satellites, which reduces the risks as another advantage. On the other hand, context-aware access control [44] should be considered as a potential approach to safeguard satellites further.Multiple satellite coordination. A constellation of satellites can multicast oracles for more extensive coverage, which can benefit space–terrestrial blockchains. Our protocol can easily expand from a single satellite setting to a constellation of participation satellites. However, it might compromise the blockchain security since more vulnerabilities are exposed in a constellation. More works need to be done for a practical space–terrestrial blockchain with multiple satellites.Information propagation model. In Section 5.2, we assess the impact of propagation delay by only considering $i_{\Delta }$, the fraction of informed nodes after one round. However, in the real world, network delays could be long-tailed or even unbounded [29]. A more general information propagation model is needed to characterize the terrestrial network and evaluate the throughput more accurately in future works.

## Figures and Tables

**Figure 1 sensors-20-05616-f001:**
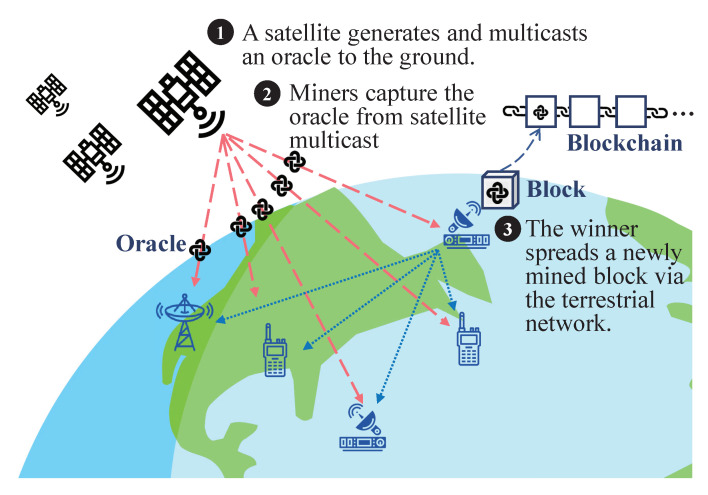
The illustration of the proposed satellite-aided consensus protocol.

**Figure 2 sensors-20-05616-f002:**
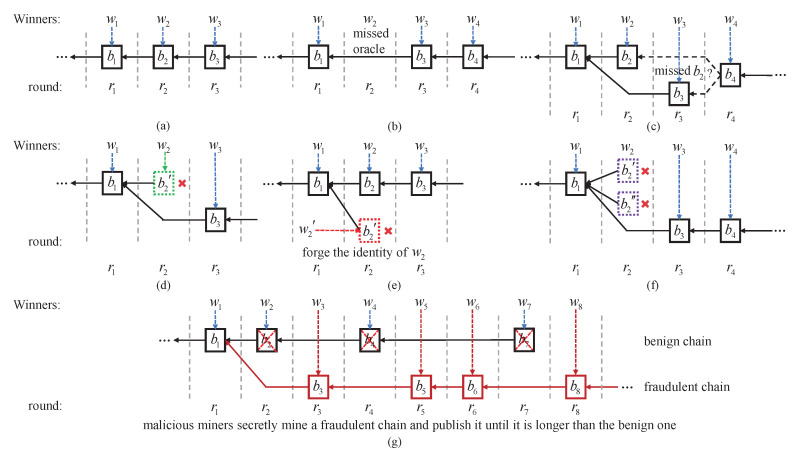
Possible patterns of blockchain evolution based on the satellite-aided consensus. (**a**) Regular extension. (**b**) Oracle missing. (**c**) Block missing. (**d**) Invalid blocks. (**e**) Fraud identity. (**f**) Multi-block in one round. (**g**) Alternative history attack.

**Figure 3 sensors-20-05616-f003:**
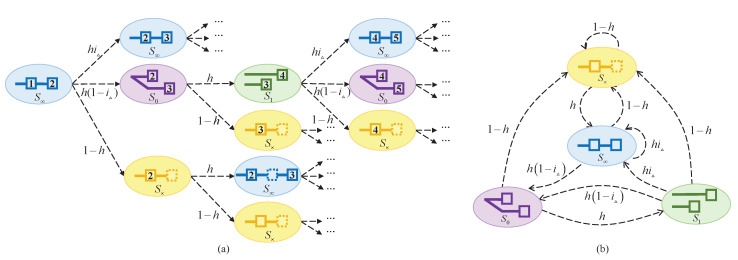
Blockchain evolution affected by information propagation delay. (**a**) The number of chain states skyrockets after three rounds. (**b**) Chain state space and transition diagram.

**Figure 4 sensors-20-05616-f004:**
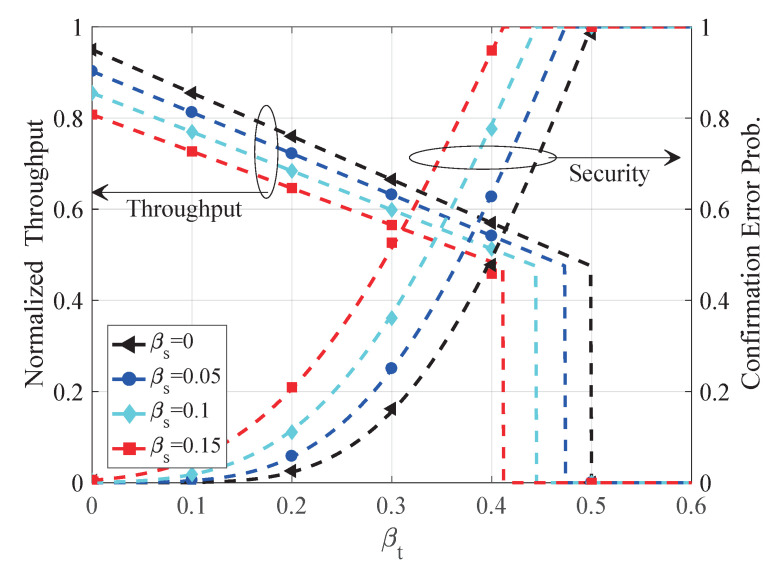
Analytical and simulation results of throughput and safety property of the proposed protocol.

**Figure 5 sensors-20-05616-f005:**
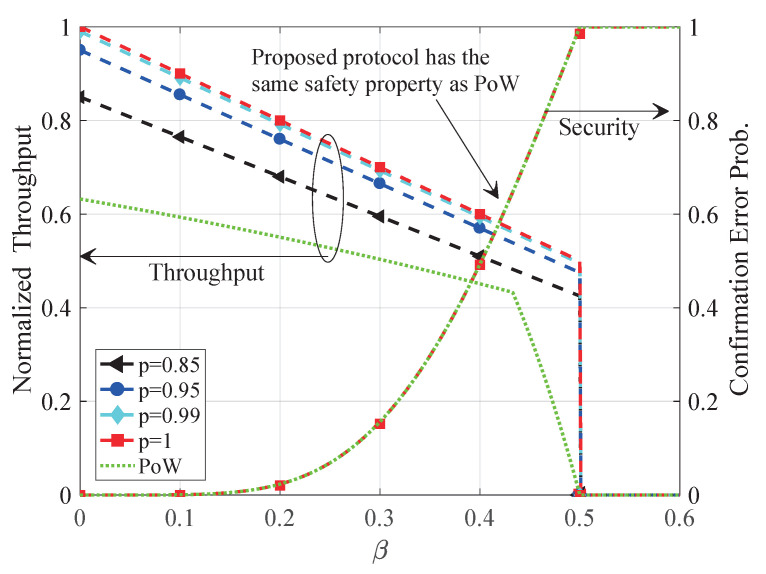
Comparison of the proposed protocol and PoW regarding throughput and safety.

**Figure 6 sensors-20-05616-f006:**
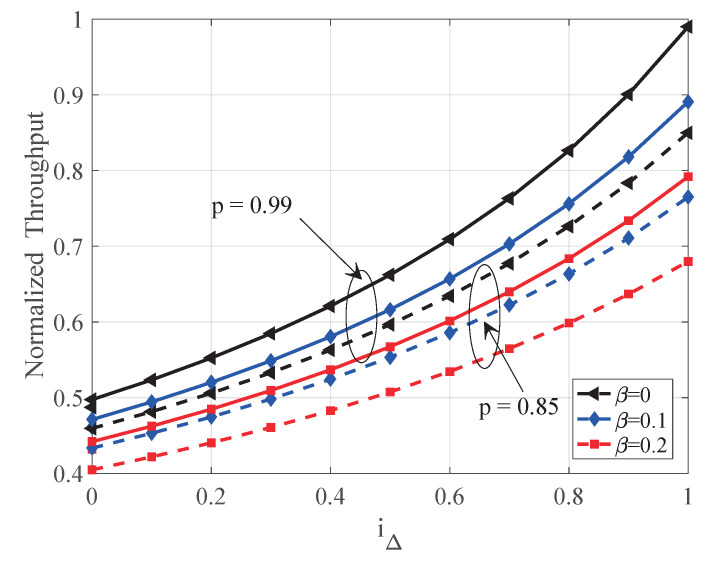
The impact of network delay on the throughput of the proposed protocol.

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
