# Peer review of "Satellite-Aided Consensus Protocol for Scalable Blockchains"

_sensors, 2020, doi:10.3390/s20195616_

Round 1

Reviewer 1 Report

This paper presents a new and interesting blockchain protocol that replaces the conventional proof-of-work mechanism for selecting the miners of a blockchain ledger with a satellite oracle. Overall, the paper is well written; the problem's formulation is sound; the theoretical analysis and simulation are adequate.

My major concern is in the necessity and feasibility of using satellites to generate the "oracles."

  1. One critical implicit assumption of the design is that the satellite has to know all the miners' identities. If the satellite doesn't know the miners' identities, a Sybil attack is possible, as the authors pointed out in line 130. But if the satellite knows the miners' identities, how does it know? In the authors’ simulation, the miners are fixed. But in realistic scenarios, the miners can change. Does the satellite have to maintain a list of live miners? If so, when new miners join the network, how does the satellite verify their legitimacy?
  2. If the miners are selected in a PoS style, why does it have to be a satellite which decides who wins, as an oracle machine can be realized by any other terrestrial devices? Actually, compared to a terrestrial device, the advantages of satellites are that they have a fixed communication delay (although the delay can be quite long, as pointed out by the authors) and that they have a large coverage (therefore shrinking the diameters of terrestrial P2P networks, as per in the design of [18] Wei et al. 2020.) Why do the oracles need to be multicast in such a way? Intuitively, the delay of an oracle (who the miner is) arriving at a blockchain network node (which can be quite short even for terrestrial oracle machines, as the oracle messages are short) does not affect it performing verification to the block generated by the selected miner. The authors are encouraged to explain the rationale behind their design more clearly and probably compare the design of using a terrestrial oracle with a satellite oracle.
  3. I can see multiple satellites in Figure 1 and that the authors also considered the situation of a fraction of adversarial satellites in Section 4. How do multiple satellites coordinate and come up with a single oracle?

There are very few satellite-enabled blockchain protocols designed so far, but the one published by your colleagues (Zhang and Liu, 2020) is probably worth noting. Their design, although quite different from the one you proposed here and probably can be considered a special case of Wei et al. 2020, provides with a complete blockchain protocol design which utilizes a satellite as the major message passing channel in the blockchain network.

Reviewer 2 Report

The contributions seem good to be published as an article. However, the following comments need to be addressed and revised the article again.

  • The key contributions should be clearly listed in the abstract. The section is currently a very short one, need more information.
  • The key findings should be listed in the conclusion along with a little background and the gaps of existing research.
  • The final section can be named as “Conclusion and Future Research”.
  • A few future research directions should be listed for researchers who are working in this direction or in the relevant directions.
  • Currently, we are connecting everyone and everything through internet of things, even the satellites are also going in this direction, like internet of satellite things, internet of healthcare things. However, such IoST/IoHT networks/assets are more susceptible to be compromised by unauthorized users. The authors can write future research directions focusing on this. How to protect/safeguard such assets against unauthorized access, especially to protect them not to making them vulnerable to access. The existing ‘access control’ can be merged with blockchain technologies to provide the solution approach. A reference paper can be considered to see case studies: A survey of context-aware access control  mechanisms  for  cloud and fog networks:  Taxonomy and open research issues, Sensors journal 2020. The multiple data sources are taken into account in this reference to reduce the throughput/overhead/latency: Achieving Security Scalability and Flexibility using Fog-Based Context-Aware Access Control, FGCS journal 2020.
  • The conclusion should be written in past form, e.g., “We illustrated the working…” “We investigated the maximum…”

Round 2

Reviewer 1 Report

The authors have made a very good response to the issues that I have raised in the previous report. My questions and concerns have been fully answered and resolved.

My final suggestion is that the responses to my comment 1 and 2 can be reflected in the manuscript. (I assume the blue texts in the revised paper are the only changes you made?) You don't have to put all the words there but a brief mention of the problems and reasons/solutions (especially regarding the comparison between satellite generated oracles and terrestrial generated ones) would be helpful, in the case that other readers would have similar questions and concerns as I do.

Author Response

We truly thank the reviewer for providing positive and insightful suggestions. We agree that other readers may have similar concerns. We thus have included the key points in comments 1 and 2 in the revised manuscript and highlighted in blue.

Again, we appreciate the time and effort from the reviewer in helping us improve the paper.

Reviewer 2 Report

The authors considered all the suggested comments and improved the paper.

Author Response

The authors would like to express their deepest thanks to the Reviewer for the time spent on reviewing the paper and for the valuable feedback.